# Adaptive Variance Inflation in Thompson Sampling: Efficiency, Safety, Robustness, and Beyond

**Feng Zhu**
Institute for Data, Systems, and Society
Massachusetts Institute of Technology
Cambridge, MA 02139
fengzhu@mit.edu

**David Simchi-Levi**
Institute for Data, Systems, and Society
Massachusetts Institute of Technology
Cambridge, MA 02139
dslevi@mit.edu

## Abstract

Thompson Sampling (TS) has emerged as a powerful algorithm for sequential decision-making, with strong empirical success and theoretical guarantees. However, it has been shown that its behavior under stringent safety and robustness criteria — such as safety of cumulative regret distribution and robustness to model mis-specification — can sometimes perform poorly. In this work, we try to address these aspects through the lens of adaptive variance inflation for Gaussian Thompson Sampling. Our one-line change introduces a time- and arm-dependent inflation factor into the sampling variance, and yields several compelling benefits. The resulting policy achieves provably worst-case optimal expected regret and worst-case optimal fast-decaying regret tail bounds, even in the presence of heavy-tailed (sub-exponential) noise or mis-specified environments. The policy is also robust to mis-specified noise variances. Beyond cumulative regret, we further demonstrate that our method ensures strong post-experiment guarantees: simple regret and estimation error per arm exhibit fast-decaying tail probabilities, contributing to more reliable and robust downstream decisions. Finally, we extend our policy to incorporate settings with unknown arm-specific variances and empirically validate the consistent performance of our approach across a range of environments.

## 1 Introduction

The stochastic multi-armed bandit (MAB) problem is a foundational framework for sequential decision-making under uncertainty, with broad applications ranging from recommendation systems [27] to clinical trials [33] and financial portfolio optimization [16]. A central challenge in MAB is balancing exploration and exploitation. Among the many algorithms proposed to address this trade-off, Thompson Sampling (TS), originally introduced in [32], has emerged as both a conceptually elegant and practically effective approach. As a Bayesian method, TS selects actions based on samples drawn from the posterior distributions over arm rewards. By naturally balancing exploration and exploitation through probabilistic sampling, TS enjoys near-optimal theoretical guarantees [19, 1, 2, 24, 3, 25, 20] and demonstrates excellent empirical performance [10, 25, 11]. Its simplicity of implementation and strong empirical results have contributed to its widespread adoption in real-world systems.

Despite its success, recent work has revealed notable limitations of Thompson Sampling, particularly regarding regret tail behavior [14, 30, 31] and statistical inference power [23, 29, 22]. Intuitively, this is because TS — like many other bandit algorithms — is primarily designed to maximize the *expected* cumulative reward *within* an experiment from an *instance-dependent* perspective. As a result, it tends to adapt quickly to a perceived optimal arm and reduce exploration prematurely. This limited exploration can lead to sub-optimal performance on other critical dimensions, such as safety under heavy-tailed risks and robustness to model mis-specification.

39th Conference on Neural Information Processing Systems (NeurIPS 2025).

While efficiency (i.e., good expected outcome) remains a central goal, safety (i.e., concentration of outcomes around the mean) and robustness (i.e., stability of both the mean and distribution with respect to hyperparameter tuning and environmental mis-specification) are equally important for practical deployment. In this work, we revisit Thompson Sampling from the perspective of efficiency, safety, and robustness, and ask:

*How can we modify Thompson Sampling to achieve efficiency, safety, and robustness, ensuring strong performance both during and after an experiment?*

## 1.1 Our contributions

Our work makes both methodological and practical contributions. We propose **Thompson Sampling with Variance Inflation (TS-VI)**, a simple yet effective extension of Gaussian Thompson Sampling that inflates the posterior sampling variance using a carefully designed time- and arm-dependent factor, with the goal of achieving strong theoretical guarantees and robust empirical performance — particularly in settings that involve multiple tasks and require safe decision-making.

**Within-experiment regret control.** We show that TS-VI achieves a worst-case optimal cumulative regret tail decay rate of $\exp(-\tilde{\Omega}(x/\sqrt{KT}))$, where $x$ denotes the regret threshold. As a corollary, its worst-case expected regret grows at the optimal rate of $\tilde{O}(\sqrt{KT})$. Moreover, we demonstrate that TS-VI is robust: it maintains these performance guarantees even under mild mis-specification of policy hyper-parameters and environmental conditions.

**Post-experiment decision quality.** We establish that TS-VI facilitates high-quality post-experiment decisions. Specifically, both the simple regret (in best-arm identification) and the per-arm estimation error (in mean reward estimation) exhibit fast tail decay and optimal expected error rates. Our analysis further implies that TS-VI promotes exploratory behavior in worst-case environments.

**Simulation validation and practical refinement.** We complement theoretical results with numerical simulations, demonstrating that TS-VI is efficient, safe, and robust across diverse environments. To handle unknown arm-specific variances in practice, we incorporate Gamma-Normal Bayesian updates into our design and show that this amendment preserves the stability and effectiveness of the policy.

## 1.2 Other Related Work

**Safety and robustness in bandit algorithms.** Recent work has increasingly focused on understanding the safety and robustness of bandit algorithms, particularly in the context of within-experiment regret. Early studies by [6, 26] showed that regret concentration typically occurs only at a polynomial rate. More recently, [4] demonstrated that bandit algorithms targeting logarithmic expected regret can be fragile: a mis-specified risk parameter (e.g., the sub-Gaussian noise level) can lead to instance-dependent regret growing polynomially in $T$. Building on this line of work, [14, 30] established that standard bandit algorithms achieving instance-dependent $\tilde{O}(1)$ regret — such as TS and its variants — can suffer from poor tail behavior: the probability of incurring large regret decays slowly with the time horizon. [30] also show that this tail performance can be significantly improved under worst-case design. Further, [31] offers a comprehensive characterization of the trade-off between expected regret and tail risk, shedding light on the intrinsic tension between efficiency and safety.

**Bandit experimental design.** There is also a growing literature on understanding the quality of post-experiment decisions in bandit settings, shifting focus from cumulative regret to final outcomes. In best-arm identification [13, 8, 17, 28, 36, 35], a commonly used performance measure is simple regret, introduced by [7], which quantifies the gap between the optimal arm's mean and the selected arm's mean. This contrasts with the probability of selecting a sub-optimal arm [5], a metric that is highly sensitive to the smallest sub-optimality gap [9] and often becomes meaningful only at large sample sizes. For mean estimation tasks [12, 29, 34, 21], it has been shown that standard bandit policies achieving instance-dependent $\tilde{O}(1)$ regret often perform poorly due to insufficient exploration of sub-optimal arms. Much of this literature has focused on minimizing the estimation error in expectation or constructing (anytime) valid confidence intervals. However, questions of decision safety and robustness in these post-experiment settings remain largely unexplored.

Before proceeding, we introduce some other notations. Throughout the paper, we use $O(\cdot)$ ($\tilde{O}(\cdot)$) and $\Omega(\cdot)$ ($\tilde{\Omega}(\cdot)$) to present upper and lower bounds on the growth rate up to constant (logarithmic) factors, and $\Theta(\cdot)$ ($\tilde{\Theta}(\cdot)$) to characterize the rate when the upper and lower bounds match up to constant (logarithmic) factors. We use $o(\cdot)$ to present strictly dominating upper bounds. In addition, for any $a, b \in \mathbb{R}$, $a \wedge b = \min\{a, b\}$ and $a \vee b = \max\{a, b\}$. For any $a \in \mathbb{R}$, $a_+ = \max\{a, 0\}$.

## 2 Model and Setup

Let the number of arms be $K$. In each time $t = 1, 2, \cdots$, the decision maker (DM) needs to decide which arm $a_t \in [K]$ should be pulled. To be more precise, let $H_t = \{a_1, r_1, \cdots, a_{t-1}, r_{t-1}\}$ be the history prior to time $t$. When $t = 1$, $H_1 = \emptyset$. In time $t$, the DM adopts a policy $\pi_t : H_t \longmapsto a_t$ that maps the history $H_t$ to an action $a_t$, where $a_t$ follows a discrete probability distribution $\pi_t(a_t|H_t)$ on $[K]$. The environment then reveals an independent reward $r_{t,a_t} = \mu_{a_t} + \epsilon_{t,a_t}$ to the DM. Here, $\mu_{a_t}$ is the mean reward of arm $a_t$, and $\epsilon_{t,a_t}$ is an independent zero-mean noise term. We assume that $\epsilon_{t,a_t}$ is $(\sigma, \nu)$-sub-exponential. That is, for any time $t$ and arm $k$,

$$\mathbb{E}\left[\exp(\lambda \epsilon_{t,k})\right] \le \exp\left(\lambda^2 \sigma^2 / 2\right), \quad \forall \lambda : |\lambda| < 1/\nu.$$

Let $\mu = (\mu_1, \cdots, \mu_K)$ be the mean vector. Let $\mu_* = \max\{\mu_1, \cdots, \mu_K\}$ be the optimal mean reward among the $K$ arms. Note that DM does not know $\mu$ at the beginning. Let $\Delta_k = \mu_* - \mu_k$ be the gap between the optimal arm and the $k$th arm. For theoretical analysis, We assume $|\Delta_k| \le 1$ (which is not necessarily known by the DM). Let $\Gamma$ be all $\mu \in \mathbb{R}^K$ such that $|\Delta_k| \le 1$. Let $n_{t,k}$ be the number of times arm $k$ has been pulled *prior to* time $t$. That is, $n_{t,k} = \sum_{s=1}^{t-1} \mathbb{1}\{a_s = k\}$. We additionally define $t_k(n)$ as the time period that arm $k$ is pulled for the $n$th time. Let $\mu_{t,k}$ be the empirical mean of arm $k$ *prior to* time $t$. That is, $\hat{\mu}_{t,k} = \sum_{s=1}^{t-1} r_s \mathbb{1}\{a_s = k\} / \sum_{s=1}^{t-1} \mathbb{1}\{a_s = k\} = \sum_{s=1}^{n_{t,k}} r_{t_k(s)} / n_{t,k}$.

### 2.1 Evaluation metric

Denote $\mathcal{E} = (\mu; \sigma, \nu)$ as the environment parameter. Fix a time horizon $T \ge K$ (which may not be known a priori by the DM). We are interested in two types of tasks — within-experiment regret control and post-experiment decision quality, illustrated as follows.

**Within-experiment regret.** Define the cumulative regret of a policy $\pi$ under the environment $\mathcal{E}$ up to time $T$ as

$$R_{\mathcal{E}}^{\pi}(T) = \sum_{t=1}^{T} (\mu_* - \mu_{a_t}) = \sum_{k=1}^{K} n_{T+1,k} \Delta_k$$

For simplicity, we write $R_{\mu}^{\pi}(T)$ instead of $R_{\mathcal{E}}^{\pi}(T)$, but we need to keep in mind that $R_{\mu}^{\pi}(T)$ is dependent on the environment profile. We are interested in studying the efficiency metric $\sup_{\mu \in \Gamma} \mathbb{E}[R_{\mu}^{\pi}(T)]$ and the safety metric $\sup_{\mu \in \Gamma} \mathbb{P}(R_{\mu}^{\pi}(T) > x)$ for large $x$, and the robustness of these two metrics with respect to mis-specified policy hyper-parameters and environment parameters (such as $(\sigma, \nu)$).

**Post-experiment decision.** We are interested in two post-experiment decisions: best arm selection and mean estimation.

*Best arm selection.* After $T$ steps, the task is to select an arm $\hat{a}_T^*$ such that $\Delta_{\hat{a}_T^*}$ is as small as possible. In particular, we are interested in studying the efficiency metric $\sup_{\mu \in \Gamma} \mathbb{E}[\Delta_{\hat{a}_T^*}]$ and the safety metric $\sup_{\mu \in \Gamma} \mathbb{P}(\Delta_{\hat{a}_T^*} > y)$, and their robustness to policy hyper-parameters and environment parameters.

*Mean estimation.* After $T$ steps, the task is to estimate the true mean of each arm such that the error is as small as possible. In particular, we are interested in studying the efficiency metric $\sup_{\mu \in \Gamma} \mathbb{E}[\|\hat{\mu}_{T+1} - \mu\|_\infty^2]$ and the safety metric $\sup_{\mu \in \Gamma} \mathbb{P}(\|\hat{\mu}_{T+1} - \mu\|_\infty^2\| > y)$, and their robustness to policy hyper-parameters and environment parameters. Here $\hat{\mu}_{T+1}$ is the estimated arm mean vector after the experiment.

**Remarks on the worst-case analysis.** The rationale for focusing on the worst-case scenario is twofold. First, worst-case analysis provides strong, uniform guarantees that hold across all environments. In particular, it allows us to investigate policy robustness with respect to both the mean reward vector $\mu$ and the broader environment profile $(\sigma, \nu)$. In contrast, instance-dependent optimal policies are often fragile, exhibiting unsafe and highly sensitive behavior in both within- and post-experiment performance [14, 30, 29].

Second, from a practical standpoint, the DM can only observe empirical quantities — namely, the cumulative reward and the sample mean of each arm:

$$\sum_{t=1}^{T} r_t = \sum_{k=1}^{K} n_{T+1,k}\mu_k + \sum_{t=1}^{T} \epsilon_{t,a_t} \quad \text{and} \quad \frac{1}{n_{t,k}}\sum_{\ell=1}^{n_{t,k}} r_{t_k(\ell)} = \mu_k + \frac{1}{n_{t,k}}\sum_{\ell=1}^{n_{t,k}} \epsilon_{t_k(s),k}.$$

The noise terms in both expressions are generally not controllable by the decision-maker. While these terms vanish in expectation, they remain significant when considering tail probabilities of high cumulative regret or large estimation error. In such cases, it is meaningful to focus on thresholds of the form $x = \Omega(\sqrt{T})$ or $y = \Omega(1/\sqrt{T})$, as any threshold below these (e.g., $x = o(\sqrt{T})$, $y = o(1/\sqrt{T})$) will typically be dominated by the noise, rendering tail bounds ineffective or uninformative.

## 2.2 Thompson Sampling with adaptive variance inflation

We present Gaussian Thompson Sampling (TS) in Algorithm 1 and introduce our one-line modification, TS with Variance Inflation (TS-VI), in Algorithm 2. Both algorithms follow a Bayesian updating framework that corresponds to placing an improper prior on each arm, $p(\mu_k) \propto 1$ (or, approximately, a weakly informative prior $\mu_k \sim \mathcal{N}(0, \sigma_*^2)$ with $\sigma_* \to \infty$), and assuming the reward distribution for arm $k$ is Gaussian: $r_t \sim \mathcal{N}(\mu_k, \sigma_0^2)$ (see, e.g., [18]). The key distinction in TS-VI (Algorithm 2) is that each posterior sample is drawn from a Gaussian distribution whose variance is inflated relative to the standard posterior variance $\sigma_0^2/n_{t,k}$ by an adaptive factor $t/(Kn_{t,k})$. Importantly, this inflation affects only the sampling distribution; the Bayesian posterior update remains unchanged.

---

**Algorithm 1** TS

1: **Input:** $\mathcal{A} = [K], \sigma_0^2$.
2: Pull each arm once.
3: **for** $t = K + 1, \cdots$ **do**
4:     For each arm $k$, draw a random sample

$$X_{t,k} \sim \mathcal{N}\left(\hat{\mu}_{t,k}, \frac{1}{n_{t,k}}\sigma_0^2\right). \quad (1)$$

5:     Take action $a_t = \arg\max_k\{X_{t,k}\}$.
6:     Collect reward $r_{t,a_t} = \mu_{a_t} + \epsilon_{t,a_t}$.
7: **end for**

---

**Algorithm 2** TS-VI

1: **Input:** $\mathcal{A} = [K], \sigma_0^2$.
2: Pull each arm once.
3: **for** $t = K + 1, \cdots$ **do**
4:     For each arm $k$, draw a random sample

$$X_{t,k} \sim \mathcal{N}\left(\hat{\mu}_{t,k}, \frac{t/K}{n_{t,k}^2}\sigma_0^2\right). \quad (2)$$

5:     Take action $a_t = \arg\max_k\{X_{t,k}\}$.
6:     Collect reward $r_{t,a_t} = \mu_{a_t} + \epsilon_{t,a_t}$.
7: **end for**

---

# 3 Within-experiment regret

In this section, we study the safety and robustness behavior of TS-VI. Our goal is to build safety guarantees for the tail distribution of cumulative regret $R_\mu^\pi(T)$. Starting from the safety result, we also build guarantees for efficiency (low expected regret) and robustness (robust regret tail distribution).

## 3.1 Main results

**Theorem 1 (Within-experiment regret)** *Fix $\sigma_0$ and $(\sigma, \nu)$. Define $M(\sigma_0; \sigma, \nu) = 1 \vee \sigma_0 \vee \frac{(\sigma \vee \nu)^2}{\sigma_0}$. There exists absolute positive constants $c$ and $C$ such that for any $x \geq c \cdot M(\sigma_0; \sigma, \nu)\sqrt{KT \ln K \ln T}$, we have*

$$\sup_{\mu \in \Gamma} \mathbb{P}\left(R_\mu^\pi(T) > x\right) \leq \exp\left(-\frac{x}{C \cdot M(\sigma_0; \sigma, \nu)\sqrt{KT}}\right).$$

We provide a brief discussion on the proof for Theorem 1 and highlight how we address the proof challenges. Without loss of generality, we assume that arm 1 is the optimal arm. The proof is based on the fact that for *any* time $t$ a sub-optimal arm $k$ is pulled, we have

$$\hat{\mu}_{t,1} + \frac{\sqrt{t/K}}{n_{t,1}}\varepsilon_{t,1} \leq \hat{\mu}_{t,k} + \frac{\sqrt{t/K}}{n_{t,k}}\varepsilon_{t,k}$$

$$\Longleftarrow \left\{ \frac{\Delta_k}{2} \leq \frac{\sum_{\ell=1}^{n_{t,k}} \epsilon_{\ell,k}}{n_{t,k}} + \frac{\sqrt{t/K}}{n_{t,k}} \varepsilon_{t,k} \right\} \bigcup \left\{ \frac{\sum_{\ell=1}^{n_{t,1}} \epsilon_{t_1(\ell),1}}{n_{t,1}} + \frac{\sqrt{t/K}}{n_{t,1}} \varepsilon_{t,1} \leq -\frac{\Delta_k}{2} \right\} \qquad (3)$$

To establish the desired bound, it suffices to control the probability of each of the two events in (3). However, there are two main challenges that render the techniques in [30, 31] insufficient for our setting.

The first challenge lies in relating $\Delta_k$ and $n_{t,k}$ to the time index $t$ and the regret threshold $x$ in a way that yields the desired tail decay dependence on $x$, $K$, and $T$ simultaneously. This is nontrivial because $n_{t,k}$ and $t$ are inherently intertwined. We address this by carefully designing a regret decomposition, showing that if the cumulative regret reaches a level $x$, then $\Delta_k$ must satisfy a precise lower bound that depends on $K$, $T$, and $t_k(n)$, and meanwhile, $n_{T+1,k}$ must be sufficiently large.

The second challenge arises from the noise term $\varepsilon_{t,k}$, which is a mean-zero random variable beyond the DM's control. This term could cause the second event in (3) to occur with non-negligible probability. To handle this, we analyze multiple values of $t$ collectively: when $n_{T+1,k}$ is large enough, there exists — with high probability — at least one time $t$ at which arm $k$ is pulled and $\varepsilon_{t,1}$ exceeds a fixed constant $\eta > 0$. However, care must be taken in selecting the range of $t$ considered — smaller $t$ leads to smaller $\sqrt{t/K}$, which weakens the tail bound. Thus, bounding the overall tail probabilities is delicate, and we defer the full technical details to the supplementary material.

### 3.2  Implications

**Efficiency.** Theorem 1 implies that the expected regret of TS-VI is $O(\sqrt{KT \ln K \ln T})$, where in $O(\cdot)$ we are hiding a constant factor. In fact, we have that

$$\sup_{\mu \in \Gamma} \mathbb{E}[R_\mu^\pi(T)] \leq 2\bar{x} + \sup_{\mu \in \Gamma} \int_{x=\bar{x}}^{+\infty} \mathbb{P}\left(R_\mu^\pi(T) > x\right) dx \leq (2c + C) M(\sigma_0; \sigma, \nu) \sqrt{KT \ln K \ln T}.$$

The expected regret bound is worst-case optimal on both $K$ and $T$ up to a logarithmic factor. We would like to point out that in contrast to most approaches taken in the literature that obtain expected regret bound, the approach we take here is to first derive regret tail bound and then yield a guarantee in expectation. While the current work is not trying to obtain the best dependence on logarithmic factors, it would be interesting to see whether and how these logarithmic factors can be removed.

**Safety.** Theorem 1 gives the optimal regret tail decaying rate. As is shown by [30] through a two-armed bandit case, for the family of policies that obtains the worst-case expected regret performance guarantee $\tilde{O}(\sqrt{T})$, the tail probability $\mathbb{P}(R_\mu^\pi(T) > x)$ cannot be decaying faster than $\exp(-x/\sqrt{T})$ for large $x$. While [30, 31] only consider UCB-like deterministic policies, our result show that the standard TS policy, as a randomized policy, can also be amended to achieve the desired optimal safety guarantee.

**Robustness.** Lastly, we would like to emphasize the robustness performance of TS-VI, which can be of great importance in practice. Note that our policy is almost parameter-free (the only input parameter is $K$ and $\sigma_0$) without assuming any knowledge to the distribution of the environment.

*Knowledge of $K$.* Typically $K$ is known to the decision-maker. If $K$ is not known a priori or not utilized in variance inflation, we can derive a worst-case guarantee similar to that in Theorem 1 with a sub-optimal dependence on $K$ (but not on $T$) — by setting $\sigma_0' = \sqrt{K}\sigma_0$, the expected regret becomes $\tilde{O}(K\sqrt{T})$. In other words, the $1/K$ factor in the variance inflation term is a "recommended" scaling parameter that makes sure the worst-case rate has an optimal dependence on $K$.

*Robustness to hyperparameter $\sigma_0$.* TS-VI is robust to mis-specified $\sigma_0$, in the sense that for any $\sigma_0$ (as long it is positive), we can always achieve the optimal regret tail decaying rate as well as the optimal regret expectation growing rate, with only constant factors affected. Apparently, if we have prior knowledge on $\sigma$ and $\nu$ (say in the sub-gaussian case we know $\sigma$ and we know that $\nu = 0$), we can set $\sigma_0 = \sigma \vee \nu$ to obtain better constant factors. In Section 5, we will examine selection of $\sigma_0$ under various situations.

*Robustness to environment mis-specification.* A feature of our result is that we consider a sub-exponential environment, where the reward tail can be heavier than that from a sub-gaussian distribution. Regardless of whether the environment is sub-gaussian or sub-exponential, TS-VI does not

require any knowledge on environment profiles, and each random sample is drawn from a Gaussian distribution (instead of a heavier-tailed distribution that caters to the environment). As we will empirically show in Section 5, standard TS and some of it variants can lead to high hidden risk in the tail region under an exponential environment, while our policy can significantly alleviate the issue.

# 4  Post-experiment decision

In this section, we study the safety and robustness behavior of TS-VI for two types of post-experiment decisions. Our goal is to build theoretical safety guarantees for the tail distribution of simple regret $\Delta_{\hat{a}_T^*}$ and estimation error $\|\hat{\mu}_{T+1} - \mu\|_\infty$. Starting from the safety result, we build guarantees for efficiency (low expected simple regret and low expected estimation error) and robustness (robust simple regret tail distribution and robust estimation tail distribution). Our analysis also provide deeper insights on the exploration behavior of TS-VI.

For any $T \geq K$, we adopt the following decisions: (a) *Best arm selection.* We select the arm that is pulled the most often in the second half of the time horizon: $\hat{a}_T^* = \arg\max_k\{n_{T+1,k} - n_{\lceil T/2\rceil+1,k}\}$. (b) *Mean estimation.* For each arm $k$, we simply take the empirical mean $\hat{\mu}_{T+1,k}$.

## 4.1  Main results

**Theorem 2 (Post-experiment best arm selection)** *Fix $\sigma_0$ and $(\sigma, \nu)$. There exists absolute constants $c_1, C_1 > 0$ such that for any $y \geq c_1 \cdot M(\sigma_0; \sigma, \nu)\sqrt{\frac{K \ln K \ln T}{T}}$, we have*

$$\sup_{\mu \in \Gamma} \mathbb{P}\left(\Delta_{\hat{a}_T^*} > y\right) \leq \exp\left(-\frac{y}{C_1 \cdot M(\sigma_0; \sigma, \nu)}\sqrt{\frac{T}{K}}\right).$$

**Theorem 3 (Post-experiment mean estimation)** *Fix $\sigma_0$ and $(\sigma, \nu)$. There exists absolute constants $c_2, C_2 > 0$ dependent only on $\sigma_0$ and $(\sigma, \nu)$ such that for any $y \geq c_2\sqrt{\frac{K}{T}}\ln^4 T$, we have*

$$\sup_{\mu \in \Gamma} \mathbb{P}\left(\|\hat{\mu}_{T+1} - \mu\|_\infty^2 > y\right) \leq \exp\left(-\sqrt{\frac{y \wedge \sqrt{y}}{C_2}\sqrt{\frac{T}{K}}}\right).$$

The proof of Theorem 2 is based on Lemma 1, which indicates that the number of times that each sub-optimal arm is pulled for $\Omega(T/K)$ times is exponentially decaying with respect to the sub-optimality gap $\Delta_k$. Intuitively, for the second half of the whole time horizon, with very low probability that TS-VI is continuously exploring or even sticking to any sub-optimal arm. Since we are selecting the arm that is pulled the most often, Theorem 2 then follows.

**Lemma 1** *There exists absolute positive constants $c_1', C_1'$ such that for any $k$ and $\Delta_k \geq c_1' \cdot M(\sigma_0; \sigma, \nu)\sqrt{\frac{K \ln K \ln T}{T}}$,*

$$\mathbb{P}\left(n_{T+1,k} - n_{\lceil \gamma T\rceil+1} > \frac{T}{2K}\right) \leq \exp\left(-\frac{\Delta_k}{C_1' \cdot M(\sigma_0; \sigma, \nu)}\sqrt{\frac{T}{K}}\right).$$

The proof of Theorem 3 is based on Lemma 2, which shows an interesting fact that TS-VI with very high probability explores each sub-optimal arm with at least $\Omega(\sqrt{T/K})$ times. Intuitively, TS-VI circumvents tail risk by inflating the variance and doing more exploration. Since the worst-case expected regret is $\tilde{O}(\sqrt{KT})$ (followed from Theorem 1), this leaves enough space for TS-VI to explore each arm on average for $\Omega(\sqrt{T/K})$ times, which leads to more estimation accuracy shown in Theorem 3.

**Lemma 2** *There exists positive constants $c_2', C_2'$ dependent only on $\sigma_0$, $(\sigma, \nu)$ such that for any $k$, we have*

$$\mathbb{P}\left(n_{T+1,k} < c_2'\sqrt{\frac{T}{K}}\right) \leq \exp\left(-\frac{1}{C_2'}\sqrt{\frac{T}{K}}\right).$$

## 4.2 Implications

**Efficiency.** Theorem 2 implies that the expected simple regret of TS-VI is $\tilde{O}(\sqrt{K/T})$, and Theorem 3 shows that the expected deviation between the empirical and true means is also $\tilde{O}(\sqrt{K/T})$. Both results follow from similar arguments as the expected regret bound in Theorem 1. These bounds are worst-case optimal in their dependence on $K$ and $T$, up to logarithmic factors (see, e.g., [5, 29]). In particular, for Theorem 3, the $\tilde{O}(1/\sqrt{T})$ rate is known to be optimal for any policy achieving expected regret $\tilde{O}(\sqrt{T})$ [29].

**Safety.** Unlike prior works (e.g., [29]) that focus solely on expected performance, Theorems 2 and 3 also provide exponential tail bounds for simple regret and estimation error, respectively, offering reliability guarantees for decision quality. These are obtained by first deriving tail bounds — similar in spirit to Theorem 1 — and then translating them into expectation bounds. It remains an open question whether the logarithmic factors in these results can be improved or removed.

**Robustness.** TS-VI also exhibits robustness to both the prior variance $\sigma_0$ and environmental mis-specification. While additional environmental knowledge may improve empirical performance, incorrect assumptions only affect constant factors, without changing the asymptotic efficiency loss or tail decay rates.

## 5 Numerical Experiments

We conduct numerical experiments on the 2-armed bandit case to illustrate the benefits brought by our policy. The mean vector is fixed as $\mu = (-\delta, \delta)$ with $\delta = 0.3$. We focus on 4 empirical metrics:

(a) expected regret vs. $t$;

(b) log tail probability of cumulative reward $< 0$ vs. $t$;

(c) mean absolute estimation error per arm vs. $t$;

(d) log tail probability of absolute estimation error $> \delta$ vs. $t$.

For each policy considered below, we collect $10^4$ trajectories. In (a) and (c), we also show the error bars (95% confidence interval) for the expected regret and the mean absolute estimation error. In the supplementary material, we also provide experiments for $\delta = 0.5$, and a 6-armed bandit case study with $\delta = 0.3, 0.5$.

### 5.1 Environments and Results

**Well-specified environment with known variances.** We first consider Gaussian environments where the noise variances are correctly specified. We consider $\sigma^2 = 2$ for both arms. Results are provided in Figure 1.

- For both TS and TS-VI, we assume the prior is $\mathcal{N}(0, 10^3)$.
- We consider the standard TS ($\sigma_0 = \sigma$), a slightly under-specified TS ($\sigma_0 = 0.9\sigma$), and a slightly over-specified TS ($\sigma_0 = 1.1\sigma$).
- For TS-VI, we consider $\sigma_0 = 0.3\sigma, 0.4\sigma, 0.5\sigma$ — we empirically find that a $\sigma_0$ being slightly less than the true $\sigma$ yields empirically stronger performance.
- We also consider the UCB policy with the bonus term $\sigma_0\sqrt{2\ln t/n}$ [15], where $\sigma_0 = 0.9\sigma$ (slightly under-specified), $\sigma_0 = 1.0\sigma$ (standard), $1.1\sigma$ (slightly over-specified).

**Mis-specified environment with known variances.** We then consider Exponential environments with Laplacian noises — that is, the probability density function is $(2b)^{-1}\exp(-|x|/b)$. We consider $b = 1$ for both arms. Note that the variance of a Laplace distribution is $2b^2$. Results are provided in Figure 2.

- For both TS and TS-VI, we assume the prior is $\mathcal{N}(0, 10^3)$.
- For TS and UCB, we treat each sample as if it is drawn from a Gaussian distribution $\mathcal{N}(0, 2)$. We consider $\sigma_0 = 0.9\sigma, 1.0\sigma, 1.1\sigma$.

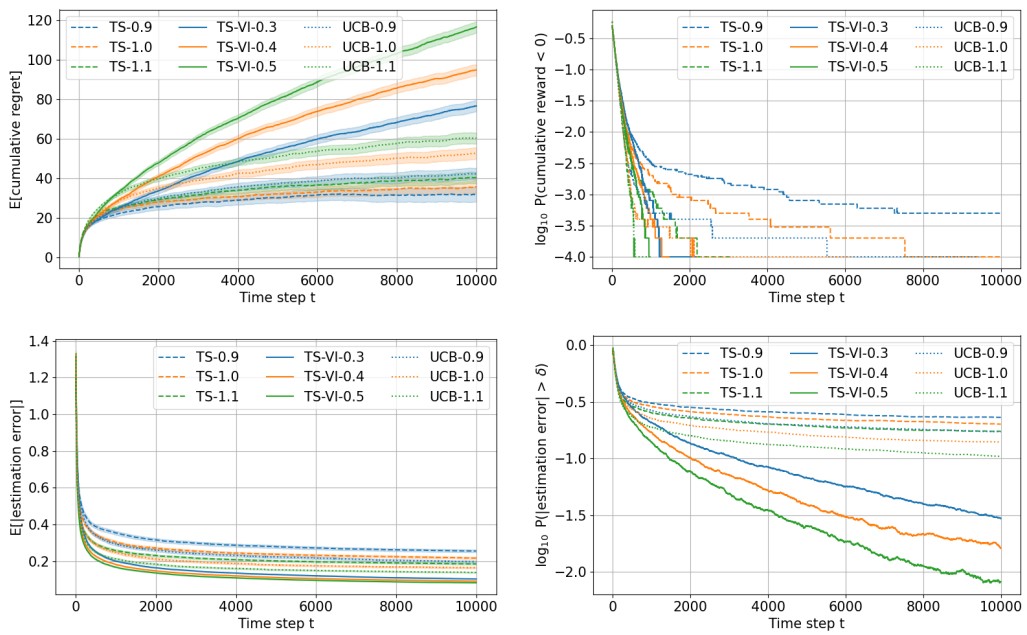

Figure 1: Results for the well-specified environment with known variances

- For TS-VI, we consider $\sigma_0 = 0.3\sigma, 0.4\sigma, 0.5\sigma$.

**Mis-specified environment with unknown heterogeneous variances.** Finally, we consider Exponential environments with noises following *unknown* Laplace distributions. We consider $b_1 = 1, b_2 = 2$. We focus exclusively on TS and TS-VI in this environment. Results are provided in Figure 3.

- Since variances are unknown, we adopt the Gamma-Normal TS where the variance of each arm $\sigma_k^2$ is modeled following an inverse Gamma distribution. The detailed paradigms of Gamma-Normal TS and its inflated version are provided in the supplementary material.

- For TS, we treat each sample as if it is drawn from a Gaussian distribution $\mathcal{N}(0, \sigma_k^2)$. The sampling variance is taken as $\sigma_{0,k} = 0.9\sigma_k, 1.0\sigma_k, 1.1\sigma_k$.

- For TS-VI, we consider $\sigma_{0,k} = 0.3\sigma_k, 0.4\sigma_k, 0.5\sigma_k$.

## 5.2 Observations and Implications

**Efficiency.** Across all environments, TS and UCB consistently exhibit lower expected within-experiment regret compared to TS-VI under various hyperparameter settings. This is expected, as TS-VI encourages greater exploration than TS, leading to more frequent pulls of sub-optimal arms — consistent with Lemma 2. However, we also observe that for smaller time horizons (e.g., $T \leq 1000$), the performance gap narrows and in some cases, TS-VI even achieves lower expected regret than TS and UCB. For larger horizons such as $T = 10000$, the increase in expected regret under TS-VI remains moderate, generally ranging from 20 to 100 (or 20 to 40 for TS-VI-0.3). This controlled sacrifice in regret is compensated by markedly improved estimation accuracy. In particular, across all settings, TS-VI consistently yields lower mean absolute estimation error compared to TS and UCB.

**Safety.** In terms of tail risk — both for within-experiment regret and post-experiment decision — TS-VI significantly outperforms TS and UCB. The empirical probability of incurring a negative cumulative reward or large estimation error for the best arm under TS-VI decays rapidly, typically approaching zero by $T \approx 5000$ (and in many cases as early as $T \approx 2000$). TS-VI also substantially mitigates the risk of large estimation errors for sub-optimal arms, whereas TS and UCB exhibits much slower tail decay.

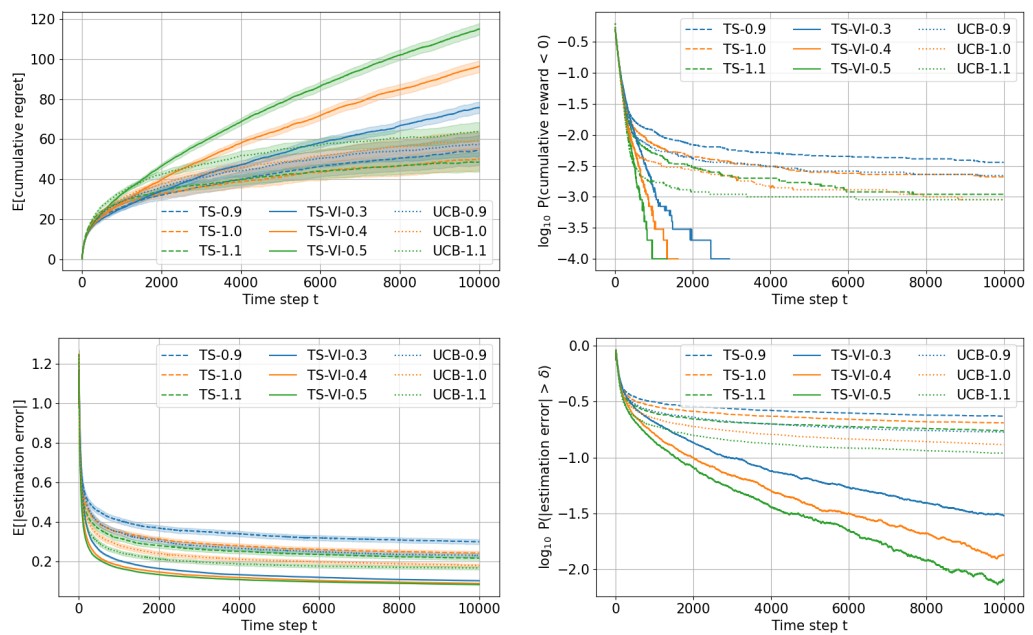

Figure 2: Results for the mis-specified environment with known variances

**Robustness.** TS-VI demonstrates stable performance across different hyperparameter settings and environmental conditions, both in terms of efficiency and safety, validating the theoretical results presented in Sections 3 and 4. In contrast, while TS reliably achieves low expected regret within an experiment, it consistently underperforms on the other three metrics: estimation accuracy, tail safety, and robustness. Notably, the safety performance of either TS or UCB deteriorates significantly in the presence of environment mis-specifications.

Finally, we would like to provide an additional remark on the performance of TS and UCB under under-specified ($\sigma_0 = 0.9\sigma$) and over-specified ($\sigma_0 = 1.1\sigma$) variances.

- For under-specified policies, while they perform well in terms of expected regret, they suffer significantly in terms of tail safety and mean estimation accuracy.

- For over-specified policies, they show slightly worse average regret but improved tail behavior and mean estimation. However, the tail decay remains slow as $T$ grows, suggesting that while empirical gains are possible via over-specification, achieving intrinsic improvement in safety necessitates explicit variance inflation. We also discuss this point via a distribution visualization in the supplementary material.

## 6 Conclusion

In this work, we propose TS-VI, a modified version of Thompson Sampling in which the sampling posterior variance is inflated by an adaptive factor. We show that TS-VI achieves efficiency, safety, and robustness in the worst-case setting, both for within-experiment regret control and post-experiment decision quality. These theoretical findings are validated through simulations, and we further extend the policy design to handle heterogeneous, arm-specific variances via Gamma-Normal Bayesian updates. Several promising directions remain for future investigation, and we highlight three below:

**Instance-dependent analysis.** While this work focuses primarily on worst-case performance, it would be insightful to conduct an instance-dependent analysis. It remains unclear whether the safety and robustness guarantees established here continue to hold under such a perspective. An instance-dependent view may offer a more nuanced understanding of how efficiency, safety, and robustness interact and trade-off with each other in different environments.

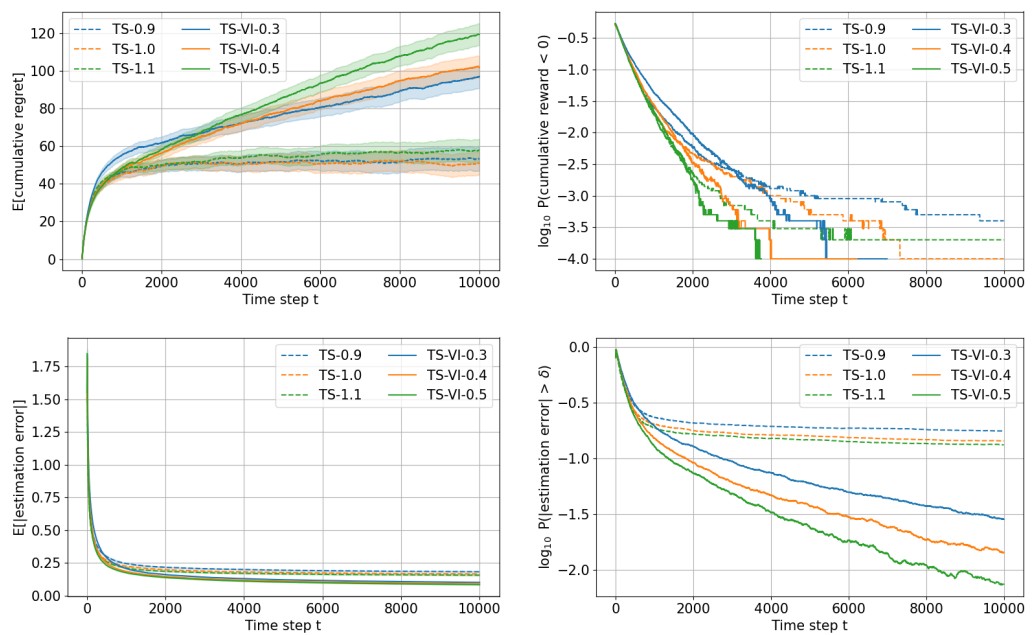

Figure 3: Results for the mis-specified environment with unknown variances

**Asymptotic behavior.** Our theoretical results provide finite-time error bounds for various within- and post-experiment objectives. However, questions remain about the asymptotic behavior of TS-VI — for example, what is the limiting proportion of times each arm is pulled? A deeper analysis of the long-run behavior could yield sharper characterizations of the policy and offer principled guidance for hyperparameter tuning.

**More complex models.** This work focuses on the standard stochastic multi-armed bandit setting. Extending our approach to more complex frameworks — such as linear bandits, nonparametric models, and Markov decision processes — is both challenging and worthwhile. It is also important to investigate how the presence of heavy-tailed reward distributions (beyond sub-exponential) affects the performance guarantees and policy behavior.

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
