# OpenReview forum: "Adaptive Variance Inflation in Thompson Sampling: Efficiency, Safety, Robustness, and Beyond"
_NeurIPS.cc/2025/Conference — NeurIPS 2025 poster_

### Official Review · Reviewer_TRXM · 2025-06-27

**Clarity:** 3
**Significance:** 3
**Originality:** 3
**Rating:** 5
**Confidence:** 3

**Summary:**

The paper analyses a modification of Thompson Sampling for Sub-exponential noise. The proposed algorithm adds a boost to the variance of the posterior samples that adapt to the noise level. The paper provides a mean $\mu$ independent, noise $(\sigma,\nu)$ dependent concentration result for the cumulative regret of their algorithm. (That does not require the knowledge of $(\mu,\sigma,\nu)$) Additionally, they provide concentration bounds of the gap between the best arm and the estimated best arm, but also on the maximal estimation error of all the arms at the end of the experiment.

**Questions:**

- Can you point me to where you show that the best action is played enough? So that $n_{t,1}$ is big enough? As it can usually be a problem for sampling algorithms.

- Would it be possible to compare in the experiment, or theoretically (because only two arms are used), this Thompson sampling algorithm to the best arm identification algorithms (with the same metrics)? So that we can see where this algorithm lies (trade-off between cumulative regret and estimation error)

- On the experiment side, I think one of the most important ones is missing. How do your performance metrics scale in $\sigma,v$? With a fixed $\sigma{0}$. I think you should also add error bars, or distribution bars, to see how the cumulative regret is distributed, as it is one of the main theoretical points of the paper. Or you may also compare the distributions of regret between the different algorithms.
This is, I think, very important, and I am willing to raise my grade if you do so.

**Ethical Concerns:**

["NO or VERY MINOR ethics concerns only"]

**Final Justification:**

The author properly addressed my concerns, and I raised my grade to 5. Provided the comparison of the cumulative regret distribution of different versions of TS is made in the main paper, and indeed shows an exponential concentration of the regret for this novel TS-VI algorithm compared to a more spread out cumulative regret for vanilla TS.

This special way of inflating the variance seems very interesting (If one is willing to trade the $log(T)$ term for a more robust algorithm). I am looking forward to seeing its expansion to other bandit settings, combinatorial, linear...

**Limitations:**

yes

**Quality:**

3

**Strengths And Weaknesses:**

Strenght

The paper is well written, and the proofs seem correct. The result on the concentration of the regret and the post estimation of the parameter is very interesting and should be explored more in general. It provides a nice addition to the analysis of sampling-like algorithms.

==============

Weakness

Some impressions should be corrected and are confusing:
- The use of $O$, $\tilde{O}$ and $\Omega$.
    - I believe that $\tilde{O}$ notation should be avoided as it can be very confusing and can hide a lot. Especially the $\tilde{O}(1)$ when talking about instant dependent regret, because it hides the dependency of the parameters. (The inverse of the gaps can be much larger than $\sqrt{T}$)
    - Line 172, I think that the $O$ notation should not leave out the variance $\sigma$ of the noise nor the $\sigma_0$ of the algorithm because it is a problem-dependent regret bound on $\sigma,\nu$ but not on $\mu$. So I think it should keep those dependencies for a better understanding of the paper.
    - Line 222 and 229,  in the $\Omega$ the $\sigma, \sigma_0, v$ should be left apparent.

- I would not exactly agree with the sentence line 30-32 : "Intuitively, this is because TS — like many other bandit algorithms — is primarily designed to maximize the expected cumulative reward within an experiment from an instance-dependent perspective.'" As far as I know, Thompson sampling was first designed for cumulative Bayesian regret, and because of that, some parameters (that have low mass according to the prior) could result in underexplored arms and then incorrect estimation.

- The number of experiments is missing, and no error intervals are shown in the experimental part. (The number of experiments appears in the checklist.)

- One of the papers cited to justify the work may state a wrong theorem 1 [4] "Bandit algorithms: Letting go of logarithmic regret for statistical robustness". Their lower bound on the regret is incorrect; a sup on the parameter (the variance) should be added in order to have infinite logarithmic normalised regret. A table of the minimum of the divergences can be found here Page 207, Bandit Algorithms
Tor Lattimore and Csaba Szepesvári from. Furthermore, in the present paper, the noise is assumed to be fixed as only a sup on $\mu$ is taken.

==============

Minor remarks

- Assuming that the gap $\Delta_{k} \leq 1$ is bounded, weakens the claim of the paper a bit. But it may provide a cleaner result for the statement of your theorems.

---

> ### Author Rebuttal · Authors · 2025-07-31
>
> 1. *The use of $O$, $\tilde O$ and $\Omega$.*
> - Thank you for the comment. We follow standard practice in using these notations to focus on dependencies with respect to $K$, $T$, and other key parameters. Where possible (e.g., Theorems 1 and 2), we explicitly specify environmental parameters and constants. We plan to provide detailed expressions --- including environmental dependencies --- in the appendix and will revise the main text accordingly.
>
> 2. *I would not exactly agree with the sentence line 30-32: "Intuitively, this is because TS — like many other bandit algorithms — is primarily designed to maximize the expected cumulative reward within an experiment from an instance-dependent perspective." As far as I know, Thompson sampling was first designed for cumulative Bayesian regret, and because of that, some parameters (that have low mass according to the prior) could result in underexplored arms and then incorrect estimation.*
> - Thank you for pointing this out. We agree that the sentence could be misleading. Our intent was to emphasize that TS, while originally motivated by Bayesian regret minimization, often lacks strong guarantees for safety and post-experiment decisions, particularly when under-exploration occurs. Even when the prior is not severely mis-specified, insufficient exploration can still lead to unreliable estimates. We will revise the sentence to reflect this more accurately.
>
> 3. *The number of experiments is missing, and no error intervals are shown in the experimental part. (The number of experiments appears in the checklist.)*
> - Thank you. As noted in the experimental section, each setting involves 10000 trials with different random seeds. We report both the mean and log tail probabilities to capture distributional characteristics. We will include standard deviations and error bars in the final version. However, we caution that standard deviation alone may not fully characterize safety, particularly in heavy-tailed settings.
>
> 4. *One of the papers cited to justify the work may state a wrong theorem 1 [4] "Bandit algorithms: Letting go of logarithmic regret for statistical robustness". Their lower bound on the regret is incorrect; a sup on the parameter (the variance) should be added in order to have infinite logarithmic normalised regret. A table of the minimum of the divergences can be found here from Page 207, Bandit Algorithms Tor Lattimore and Csaba Szepesvári. Furthermore, in the present paper, the noise is assumed to be fixed as only a sup on $\mu$ is taken.*
> - Thank you for pointing this out. We reference [4] to highlight robustness limitations in standard UCB and TS under volatility mis-specification. We will clarify that our citation is intended to support this point, and not necessarily to endorse every claim in [4]. We will also correct the reference and note its limitations in the final version.
>
> 5. *Assuming that the gap $|\Delta_k|\leq 1$ is bounded, weakens the claim of the paper a bit. But it may provide a cleaner result for the statement of your theorems.*
> - We agree. The boundedness assumption is made to simplify theoretical presentation. Our algorithm does not require prior knowledge of the bound. If $|\Delta_k| \leq C$, our regret bounds scale proportionally with $C$, and tail bounds adjust accordingly (an additional factor of $C$ on the denominator in the exponents). We will clarify this in the final version.
>
> 6. *Can you point me to where you show that the best action is played enough, so that $n_{t, 1}$ is big enough? As it can usually be a problem for sampling algorithms.*
> - Theorem 1 and its proof demonstrate that the best arm is played at least $T - O(\sqrt{T}/\Delta)$ times with high probability, where $\Delta$ is the minimum gap. When $\Delta$ is small, arm pulls may become more balanced --- this is expected behavior and consistent with existing lower bounds.
>
> 7. *Would it be possible to compare in the experiment, or theoretically (because only two arms are used), this Thompson sampling algorithm to the best arm identification algorithms (with the same metrics)? So that we can see where this algorithm lies (trade-off between cumulative regret and estimation error)?*
> - Thank you for the suggestion. Due to space constraints, detailed comparisons with best-arm identification algorithms are not in the main paper, but multi-arm results are provided in the appendix. Previous work (e.g., [28]) has explored the trade-off between cumulative regret and estimation error. Our work offers a unified framework that achieves worst-case optimal performance for both, and we are happy to expand on this point in the final version and pursue direct comparisons in future work.
>
> 8. *On the experiment side, I think one of the most important ones is missing. How do your performance metrics scale in $\sigma, \nu$, with a fixed $\sigma_0$? I think you should also add error bars, or distribution bars, to see how the cumulative regret is distributed, as it is one of the main theoretical points of the paper. Or you may also compare the distributions of regret between the different algorithms. This is, I think, very important, and I am willing to raise my grade if you do so.*
>
> - We appreciate your detailed comment. Our experiments (including those in the appendix) illustrate that TS-VI-$\eta$ with $\eta=0.3$ to $0.6$ appears robust and safe under various different environments, while retaining good efficiency. This corresponds to fixing $\sigma_0$ while varying $\sigma$ and $\nu$. We are willing to include a direct comparison among those environments for fixed $\eta$ in the final version. In our experiments, for each environment and each policy, we run 10000 trials with different random seeds. We report both mean and log tail probabilities (serve as an analogous functionality as standard deviations). We will report standard deviations in the final version of our paper. Meanwhile, we would also like to point out that standard deviation might not fully capture the safety of a policy because there might be some chance that the tail is not characterized by the standard deviation. Initially we did not incorporate comparison of distributions because there are so many policies and various $T$. In the final version, we will plot comparison of distributions when $T=10000$ for various environments.

---

> > ### Comment · Reviewer_TRXM · 2025-08-04
> >
> > The author properly addressed my concerns, and I raised my grade to 5. Provided the comparison of the cumulative regret distribution of different versions of TS is made in the main paper, and indeed shows an exponential concentration of the regret for this novel TS-VI algorithm compared to a more spread out cumulative regret for vanilla TS.
> >
> > This special way of inflating the variance seems very interesting. (If one is willing to trade the $\log(t)$ term for a more robust algorithm).

---

> > > ### Author Response · Authors · 2025-08-06
> > >
> > > Thank you very much for the comment and your appreciation on the variance inflation factor. We will add experiments,  distributional plots, and error bars accordingly in the final version.

---

### Official Review · Reviewer_exTB · 2025-06-28

**Clarity:** 3
**Significance:** 2
**Originality:** 2
**Rating:** 4
**Confidence:** 3

**Summary:**

This paper proposes a modification to Thompson Sampling by introducing a time- and arm-dependent inflation factor into the sample variance. The authors establish strong theoretical guarantees for the proposed policy, demonstrating that it achieves worst-case optimal expected regret as well as a worst-case optimal, exponentially decaying regret tail bound. Furthermore, the paper provides post-experimental theoretical guarantees to complement the proposed method.

**Questions:**

See Weaknesses.

**Ethical Concerns:**

["NO or VERY MINOR ethics concerns only"]

**Final Justification:**

The authors have addressed my concerns. Accordingly, I increase my score.

**Limitations:**

See Weaknesses.

**Paper Formatting Concerns:**

It would improve the clarity of the discussion in lines 287--291 if it were explicitly linked to the corresponding figures.

**Quality:**

2

**Strengths And Weaknesses:**

Strengths
- The proposed method is conceptually simple and easy to understand, while being supported by solid and rigorous theoretical guarantees.

Weaknesses
- The intuition behind the adaptive inflation factor $t / (K n_{t,k})$ is not entirely clear. Is there a principled criterion or theoretical justification for choosing this form? More broadly, if one introduces an unknown adaptive factor, it seems possible to search for other forms that satisfy similar efficiency requirements.

- Beyond identifying the optimal mean reward, can the proposed method be extended to settings where the goal is to estimate the difference in means between arms? Such a scenario is of particular practical interest.

- The experimental evaluation raises several concerns:
     - What happens if the hyperparameter $\sigma_0$ is set equal to or larger than the true $\sigma$?
     - It would be beneficial to include experiments with $K > 2$ arms, as well as scenarios where $K$ is unknown to the learner.
     - From Figures 1--3, the proposed method appears to be sensitive to the choice of the hyperparameter $\sigma_0$, with different values leading to noticeably different performance. This raises concerns about the robustness of the approach.
    - $\sigma_0$ also appears in standard Thompson Sampling (TS). It would be helpful to report the performance of TS when $\sigma_0$ is slightly smaller than the true $\sigma$, for a more direct comparison with the proposed method.

---

> ### Author Rebuttal · Authors · 2025-07-31
>
> 1. *The intuition behind the adaptive inflation factor $t / (K n_{t,k})$ is not entirely clear. Is there a principled criterion or theoretical justification for choosing this form? More broadly, if one introduces an unknown adaptive factor, it seems possible to search for other forms that satisfy similar efficiency requirements.*
> - Thank you for this excellent question. Indeed, several inflation forms could satisfy the same regret expectations—for example, $(t/Kn)^\alpha$ for $\alpha \in (0, 1/2]$. However, only $\alpha = 1/2$ ensures the optimal regret tail decay rate for any $x > 0$, given the worst-case expected regret of $O(\sqrt{KT})$. The linear scaling with $n$ and square-root scaling with $t$ are essential; deviating from these would either violate worst-case optimal regret or degrade tail performance.
>
> 2. *Beyond identifying the optimal mean reward, can the proposed method be extended to settings where the goal is to estimate the difference in means between arms? Such a scenario is of particular practical interest.*
> - Yes, our method extends naturally to estimating differences in means between arms. Theorem 3 implies controlled estimation error for each arm. Specifically, Lemma 2 shows that each arm is sampled at least $\Theta(T/K)$ times with high probability, ensuring reliable estimates of both individual means and their differences.
>
> 3. *What happens if the hyperparameter $\sigma_0$ is set equal to or larger than the true $\sigma$?*
> - Thank you for pointing out this ambiguity. Our focus is on underestimation of $\sigma$, where the decision-maker risks being overly aggressive due to under-perceived uncertainty. If the hyperparameter equals or exceeds the true $\sigma$, the policy typically maintains optimal expected regret, although practical performance may degrade due to conservativeness. However, tail behavior remains polynomial, as shown in Simchi-Levi et al. (2022). Moreover, if the environment is mis-specified (e.g., exponential rather than Gaussian), even a larger $\sigma_0$ may not suffice --- highlighting the need for robust policies.
>
> 4. *It would be beneficial to include experiments with $K > 2$ arms, as well as scenarios where $K$ is unknown to the learner.*
> - Multi-arm experiments are included in the appendix due to space constraints. As for unknown $K$, our current policy assumes knowledge of $K$, and extending it to handle unknown $K$ remains an open direction for future work.
>
> 5. *From Figures 1-3, the proposed method appears to be sensitive to the choice of the hyperparameter $\sigma_0$, with different values leading to noticeably different performance. This raises concerns about the robustness of the approach.*
> - We appreciate the concern. In our context, ``robustness'' refers specifically to protection against under-specification of risk. As shown in Figures 1-3, while regret expectations do vary with the hyperparameter, the regret tail decays much more favorably with our method. This trade-off is inherent—some loss in expected performance is necessary to gain safety. That said, improving robustness in expectation (i.e., reducing constant-factor differences) is an interesting direction for future work.
>
> 6. *$\sigma_0$ also appears in standard Thompson Sampling (TS). It would be helpful to report the performance of TS when $\sigma_0$ is slightly smaller than the true $\sigma$, for a more direct comparison with the proposed method.*
> - Thank you for the suggestion. We have included experiments with standard TS using a slightly mis-specified variance ($(0.9\sigma)^2$), along with UCB baselines: standard UCB ($\sigma\sqrt{2\ln t/n}$), and UCB with mis-specified bonus ($0.9\sigma\sqrt{2\ln t/n}$). Our conclusions remain the same: while these policies perform well in terms of average regret, they exhibit poor tail performance and mean estimation quality. These results will be added to the final version. We reiterate that our TS-GN policy offers a more robust alternative under uncertainty and heterogeneity.

---

> > ### Comment · Reviewer_exTB · 2025-08-03
> >
> > I thank the authors for their efforts. Most of my concerns have been addressed. However, I had expected additional experiments where the hyperparameter $\sigma_0$ is set equal to or larger than the true $\sigma$, which were not included in the response. It would also be helpful to include a discussion of additional experiments in the main text.

---

> > > ### Author Response · Authors · 2025-08-06
> > >
> > > Thank you very much for the response. We did not realize that the over-misspecification helps improve the completeness of the paper, and thank you for pointing this out. We have conducted experiments with standard TS using a slightly over-specified variance ($(1.1\sigma)^2$), along with UCB with over-specified bonus ($1.1\sigma\sqrt{2\ln t/n}$). We find that these policies perform slightly worse in terms of average regret, they exhibit slightly better tail performance and mean estimation quality --- but still the tail decaying rate is polynomial. So the conclusion is --- empirically over-misspecification helps, but to get intrinsic improvement one need to resort to variance inflation. We will include the experiments in the final version.

---

> > > > ### Comment · Reviewer_exTB · 2025-08-06
> > > >
> > > > Thank you for the additional efforts. Including further discussion on over-misspecification in the final version would be helpful. I will raise my score.

---

### Official Review · Reviewer_3uyB · 2025-06-30

**Clarity:** 2
**Significance:** 3
**Originality:** 3
**Rating:** 4
**Confidence:** 3

**Summary:**

This paper proposes Thompson Sampling with Variance Inflation (TS-VI), a modification of Gaussian TS that inflates the posterior sampling variance by a time- and arm-dependent factor `t/(Kn)`. The key contribution is achieving simultaneously optimal expected regret, optimal regret tail decay, and strong post-experiment performance for both best-arm identification and mean estimation tasks. The authors provide theoretical guarantees showing that TS-VI maintains robustness to hyperparameter misspecification and environmental conditions, including heavy-tailed (sub-exponential) noise. The approach is validated through numerical experiments on multi-armed bandit problems under various noise conditions.

**Questions:**

1. Experimental validation lacks uncertainty quantification. Repeating trials with multiple random seeds and reporting standard deviations would strengthen the empirical claims.

2. The experimental section would benefit from comparison with other baselines beyond standard TS, such as UCB or $\epsilon$-TS [1].

3. What would be the main challenges in extending the variance inflation idea to a more general MDP setting? How could this work benefit MDP analysis?

**Reference:**
[1] Thompson Sampling with Less Exploration is Fast and Optimal, ICML 2023

**Ethical Concerns:**

["NO or VERY MINOR ethics concerns only"]

**Final Justification:**

The authors have adequately addressed my main concerns. However, as I mentioned earlier, I find it difficult to assess the significance of the contribution in terms of safety and robustness. The paper and the rebuttal do not make this aspect of the work particularly clear to me. Therefore, I will maintain my current score with low confidence.

**Limitations:**

The *mistake in Theorem 3* (lines 812-815 in the appendix) should be corrected in the final version.

The work does not raise immediate ethical concerns.

**Paper Formatting Concerns:**

There is no major formatting issue. For other formatting concerns, please check *Strengths and Weaknesses*.

**Quality:**

2

**Strengths And Weaknesses:**

1. **Quality:**
The theoretical analysis is comprehensive and technically solid, but the manuscript suffers from numerous notation inconsistencies and mathematical presentation issues that significantly affect readability. The writing feels somewhat hasty, and the paper would benefit from a careful revision to improve clarity and rigor in notation. For example:

    * Line 102: `r_{t_k(n)} / n_{t,k}` should be `r_{t_k(s)} / n_{t,k}` (inconsistent summation index).
    * Lines 115–118: Inconsistent notation between `\mu^*` and `\mu_*` (as used in line 96); the description “approximating the true optimal arm `\mu^*`” is conceptually unclear.
    * Line 121: `\hat{\mu}_{T+1}` lacks an arm subscript, and the expression  `\left\|\hat{\mu}_{T+1} - \mu\right\|_\infty^2 > y`
      contains an extra delimiter in the original version.
    * Line 214: `\hat{a}^{*}_t` should be `\hat{a}^{*}_T`.
    * Algorithms 1–2: The loop should use `for t = 1 to T` instead of starting from `K+1`. Since the only difference between TS and TS-VI is the variance term, the two algorithms could be merged into one with a parameter to control the variance setting.

 2. **Clarity:**
   Overall, the paper is readable, and the regret analysis appears reasonable. Adaptive variance inflation forces each arm to explore suboptimal arms more and to be sampled at least `\Omega(\sqrt{T/K})` times. *The reviewer acknowledges a limited prior background in the safety and robustness literature and is therefore not fully confident in evaluating the appropriateness of the analysis (section 4) from that perspective.*

    Some conceptual descriptions are imprecise and impact readability. For example, in the discussion of post-experiment evaluation metrics (lines 115–118), the best-arm selection task is described as choosing “an arm `\hat{a}^T` such that `\mu_{\hat{a}^T}` approximates the true optimal arm `\mu^*` as close as possible.” This phrasing is conceptually unclear, as both `\mu_{\hat{a}^T}` and `\mu^*` refer to true mean values.

    It is recommended that the authors carefully review the paper and revise such language to improve conceptual clarity and ensure the presentation is precise throughout.

3. **Significance:**
This work makes a meaningful contribution to our understanding of safety and robustness in TS, which has been underexplored despite TS's widespread practical adoption. The unified framework that addresses both cumulative regret and post-experiment decision quality is especially valuable for practical deployment.

4. **Originality:**
The proposed variance inflation mechanism is novel and conceptually simple. The accompanying theoretical framework, which unifies efficiency, safety, and robustness through worst-case and tail analyses, offers new insights for the design of safe bandit algorithms.

---

> ### Author Rebuttal · Authors · 2025-07-31
>
> We would like to first thank the reviewer for the very careful correction on some typos in the paper. We will make corresponding amendment, revise such language, and correct all mistakes in the final version.
> 1. *Experimental validation lacks uncertainty quantification. Repeating trials with multiple random seeds and reporting standard deviations would strengthen the empirical claims.*
> - Thank you for the suggestion. As stated in the experimental section, we conduct 10,000 trials for each environment-policy pair using different random seeds. We report both the mean and the log tail probabilities, which serve a similar purpose as standard deviations by capturing distributional behavior. We will explicitly include standard deviations in the final version. However, we note that standard deviations may not fully characterize safety, particularly when tail events deviate from Gaussian assumptions.
>
> 2. *The experimental section would benefit from comparison with other baselines beyond standard TS, such as UCB or $\epsilon$-TS [1].*
> - Thank you for the suggestion. In addition to standard TS, we also evaluated standard TS with slightly mis-specified variance ($(0.9\sigma)^2$), standard UCB ($\sigma\sqrt{2\ln t/n}$), and UCB with mis-specified bonus ($0.9\sigma\sqrt{2\ln t/n}$). Our findings hold across these baselines—while these methods perform well in terms of expected reward, they suffer significantly in terms of tail safety and mean estimation. These additional results will be included in the final version. We again emphasize that TS-GN is a practical solution for handling unknown heterogeneous variances across arms.
>
> 3. *What would be the main challenges in extending the variance inflation idea to a more general MDP setting? How could this work benefit MDP analysis?*
> - We assume the reviewer is referring to the episodic MDP setting. The main challenge is that current actions affect future states, leading to error propagation over time. Estimation errors in later stages can also influence earlier decisions, making analysis much more complex. While we have begun exploring extensions of variance inflation to MDPs, establishing theoretical guarantees appears to require new tools beyond our current variance inflation framework.

---

> ### Comment · Reviewer_3uyB · 2025-08-05
>
> Thank you for the response. My main concerns have been addressed. I will maintain my original score, as I am unsure whether the work meets the acceptance bar, in part due to my limited familiarity with the safety and robustness literature.
>
> I also pointed out some notational issues under "Quality and Clarity". Please revise accordingly if appropriate.

---

> > ### Author Response · Authors · 2025-08-06
> >
> > Thank you for your response, and we hope our clarification helps emphasize that our work improves the practicality of bandit algorithms under volatile environments by incorporating safety and robustness. We will modify all the notational issues you pointed out. Thanks again for the detailed correction.

---

### Official Review · Reviewer_vVC8 · 2025-07-08

**Clarity:** 3
**Significance:** 2
**Originality:** 2
**Rating:** 4
**Confidence:** 2

**Summary:**

This paper proposes a modification of the well-known Thompson Sampling method to balance efficiency, safety, and robustness both during and after an online learning experiment in the multi-armed bandit setting. The authors consider three key evaluation scenarios for these metrics: i) within-experiment cumulative regret $R\_T$, ii) post-experiment best-arm selection regret $\Delta\_{\hat{a}\_T}$, and iii) post-experiment mean reward estimation $\hat{\mu}\_{T+1}$. The proposed method, TS with Variance Inflation (TS-VI), differs from standard Gaussian TS through a slight modification of the variance term in the sampling step, designed to encourage greater exploration.

Rather than deriving bounds on the expected values of these metrics for specific environmental hyperparameter settings (eg, subexponential noise parameters and reward means), the authors first establish (in Theorems 1, 2, and 3) uniform upper bounds on the tail probabilities of these metrics across a set of mean rewards, i.e., $\sup_{\mu \in \Theta}\mathbb{P}(\texttt{metric} > x)$. These bounds serve as a proxy for safety and, in turn, enable the derivation of bounds on the expected values of the metrics, serving as a proxy for efficiency. The authors further argue that the asymptotic bounds are robust to misspecification of environmental parameters, as their effect appear only as constant factors.

**Questions:**

1. What exactly is the set $\Theta$?

2. What is the precise dependence of the constant factors in your bounds on the environmental hyperparameters?

**Ethical Concerns:**

["NO or VERY MINOR ethics concerns only"]

**Final Justification:**

Rebuttal addressed some of my questions. I keep my score the same because I cannot assess the paper properly given that I am not an expert in this field and unfamiliar with the literature. So my assessment does not cary as much weight.

**Limitations:**

Yes

**Paper Formatting Concerns:**

non

**Quality:**

3

**Strengths And Weaknesses:**

Strengths:
1. The proposed algorithm is a straightforward modification of Thompson Sampling and can be easily implemented.
2. The uniform tail probability bounds on within-experiment regret and post-experiment decision quality provide a more comprehensive framework for analyzing the performance of such algorithms from multiple perspectives (e.g., safety, robustness, and efficiency).
3. The results are clearly stated and well organized.

Weaknesses:
1. The significance of the results is not well motivated. As someone not directly working in this field, I find it difficult to appreciate the broader contributions of this paper. It is also unclear which aspects of the work represent novel contributions to the literature.
2. While I understand this is primarily a theoretical paper, I would appreciate more extensive empirical studies to support the theoretical claims and help make the insights more tangible. The current version, including the experiments in the appendix, focuses solely on synthetic scenarios. The impact of the paper could be significantly elevated by including empirical studies involving real-world problems and drawing insights from them to demonstrate the practical relevance of the results.

---

> ### Author Rebuttal · Authors · 2025-07-31
>
> 1. *The significance of the results is not well motivated. As someone not directly working in this field, I find it difficult to appreciate the broader contributions of this paper. It is also unclear which aspects of the work represent novel contributions to the literature.*
> - Thank you for the insightful comment. We would like to clarify our contributions more explicitly. (a) On the practical side, our work aims to develop bandit policies that simultaneously achieve optimal expected regret, optimal tail decay in regret, and strong post-experiment performance in both best-arm identification and mean estimation tasks. In real-world scenarios, decision-makers must balance multiple objectives—accounting not just for within-experiment performance, but also for the downstream impact of decisions made under uncertainty and possible model mis-specification. (b) On the theoretical side, our work proposes a conceptually simple yet powerful modification to the Thompson Sampling (TS) policy, along with a unifying analysis framework that ensures efficiency, safety, and robustness across multiple objectives. We believe this framework offers fresh insights for the design of safe and practically reliable bandit algorithms.
>
> 2. *While I understand this is primarily a theoretical paper, I would appreciate more extensive empirical studies to support the theoretical claims and help make the insights more tangible. The current version, including the experiments in the appendix, focuses solely on synthetic scenarios. The impact of the paper could be significantly elevated by including empirical studies involving real-world problems and drawing insights from them to demonstrate the practical relevance of the results.*
> - Thank you for the valuable suggestion. We agree that incorporating real-world experiments would enhance the practical impact of our work, and we plan to explore this in future work. Nonetheless, we wish to emphasize that our proposed TS-GN policy is already a practical enhancement --- it specifically addresses the challenge of unknown heterogeneous variance across arms, which is common in applied settings.
>
> 3. *What exactly is the set $\Theta$?*
> - $\Theta$ is the set of all vectors in $\mathbb R^d$ such that $\Delta_k\leq 1$ for all $k$. We will make this more precise in the final version.
>
> 4. *What is the precise dependence of the constant factors in your bounds on the environmental hyperparameters?*
> - The dependence on environmental hyperparameters is explicitly provided in Theorems 1 and 2. For Theorem 3, the dependence is more intricate and is detailed in the appendix. While we suppress absolute constants for clarity (as they are difficult to characterize precisely), we note that loose upper bounds are available and will be clarified in the final version.

---

> > ### Comment · Reviewer_vVC8 · 2025-08-03
> >
> > I thank the authors for their detailed and thoughtful rebuttal. While I appreciate their clarifications and willingness to revise their work, I am inclined to maintain my original score, as I am still unable to fully justify whether the paper meets the accept threshold, which demands "high impact on at least one sub-area of AI or moderate-to-high impact on more than one area of AI".

---

> > > ### Author Response · Authors · 2025-08-06
> > >
> > > Thank you for the comment. We hope you find our response helpful. As for the contribution, we hope to emphasize that our contribution is both methodological and practical, in the sense that we hope to provide practical and simple modifications on standard bandit policies to achieve strong theoretical guarantees and empirical performance, taking into account practical concerns of conducting multiple tasks and making safe decisions. We will make this point clearer in the final version.

---

### Note · Authors · 2025-08-15

Dear AC and reviewers,

We would like to take this opportunity to further highlight our contributions and outline the additional experiments we will include in the final version of our paper.

### 1. Contributions

Our work makes both **methodological** and **practical** contributions. We propose simple yet effective modifications to standard bandit policies, with the goal of achieving strong theoretical guarantees and robust empirical performance—particularly in settings that involve multiple tasks and require safe decision-making.

- **Practical side:**
  We design bandit algorithms that simultaneously achieve:
  - Optimal expected regret
  - Optimal tail decay of regret
  - Strong post-experiment performance in best-arm identification and mean estimation tasks

  These criteria reflect real-world needs, where decision-makers must weigh not only within-experiment performance but also downstream consequences under uncertainty and potential model misspecification.

- **Theoretical side:**
  We present a conceptually simple but powerful modification to **Thompson Sampling (TS)**, along with a **unifying analytical framework**. This framework ensures efficiency, safety, and robustness across multiple objectives, and we believe it offers valuable new insights into the design of practically reliable and safe bandit algorithms.

### 2. Experiments

We will add plots of regret distributions when $T=10000$ and error bars of expected regret for $T\in[1, 10000]$. Furthermore, we will include:

- TS with slightly **under-specified variance** ($(0.9\sigma)^2$) and **over-specified variance** ($(1.1\sigma)^2$)
- **Standard UCB** ($\sigma\sqrt{2\ln t / n}$)
- **UCB with under-specified** ($0.9\sigma\sqrt{2\ln t / n}$) and **over-specified bonuses** ($1.1\sigma\sqrt{2\ln t / n}$)

### Key findings:

- **Under-specified case:**
  While these methods perform well in terms of expected regret, they suffer significantly in terms of tail safety and mean estimation accuracy.

- **Over-specified case:**
  These policies show slightly worse average regret but improved tail behavior and mean estimation. However, the tail decay remains polynomial, suggesting that while empirical gains are possible via over-specification, achieving intrinsic improvement in safety necessitates explicit variance inflation.

In addition to highlighting the two main issues above, we will incorporate all other modifications suggested by the reviewers.

---

### Decision · Program_Chairs · 2025-09-17

**Decision:**

Accept (poster)

**Comment:**

This paper proposes a modification to Gaussian Thompson Sampling (TS) by introducing an adaptive, time- and arm-dependent variance inflation factor. The resulting algorithm (TS-VI) is designed to balance efficiency, safety, and robustness in multi-armed bandits. The authors provide theoretical guarantees: worst-case optimal expected regret, exponentially decaying regret tails under sub-exponential noise, and robust behavior under variance misspecification. Beyond cumulative regret, the framework also ensures fast-decaying tails for simple regret and mean estimation error, making it attractive for both online and post-experiment performance. Empirical results on synthetic bandit problems, and extensions to scenarios with unknown variances.

The reviewers found the contribution technically substantial and conceptually simple, appreciating the unification of efficiency, safety, and robustness perspectives. However, multiple concerns were raised: (i) lack of clarity and notational inconsistencies, which sometimes affected readability (R3uyB, TRXM); (ii) limited experimental validation beyond synthetic scenarios, with requests for comparisons against broader baselines such as UCB and ε-TS (R3uyB, exTB); (iii) unclear intuition for the exact form of the variance inflation factor (RexTB); and (iv) insufficient discussion of practical significance for broader applications (RvVC8). The authors addressed many of these concerns in rebuttal, promising corrections to notational issues, distributional plots, additional experiments with mis-specified baselines.

Overall, I recommend acceptance. The method is novel, technically correct, and supported by meaningful theoretical advances. While empirical evaluation could be further expanded, the promised additional results and clarifications will help increase the strength of the paper. Please incorporate the final suggestions of the reviewers in the camera-ready version as promised.